# Tumor-Associated Exosomes: A Potential Therapeutic Target for Restoring Anti-Tumor T Cell Responses in Human Tumor Microenvironments

**DOI:** 10.3390/cells10113155

**Published:** 2021-11-13

**Authors:** Gautam N. Shenoy, Maulasri Bhatta, Richard B. Bankert

**Affiliations:** 1Department of Microbiology and Immunology, Jacobs School of Medicine and Biomedical Sciences, University at Buffalo, Buffalo, NY 14203, USA; gautamsh@buffalo.edu; 2Department of Immunology, Roswell Park Comprehensive Cancer Center, Buffalo, NY 14203, USA; maulasri.bhatta@roswellpark.org

**Keywords:** exosomes, T cell responses, tumor microenvironment, phosphatidylserine, PD-L1, immune checkpoints, ganglioside GD3, miRNA, FasL, xenograft models

## Abstract

Exosomes are a subset of extracellular vesicles (EVs) that are released by cells and play a variety of physiological roles including regulation of the immune system. Exosomes are heterogeneous and present in vast numbers in tumor microenvironments. A large subset of these vesicles has been demonstrated to be immunosuppressive. In this review, we focus on the suppression of T cell function by exosomes in human tumor microenvironments. We start with a brief introduction to exosomes, with emphasis on their biogenesis, isolation and characterization. Next, we discuss the immunosuppressive effect of exosomes on T cells, reviewing in vitro studies demonstrating the role of different proteins, nucleic acids and lipids known to be associated with exosome-mediated suppression of T cell function. Here, we also discuss initial proof-of-principle studies that established the potential for rescuing T cell function by blocking or targeting exosomes. In the final section, we review different in vivo models that were utilized to study as well as target exosome-mediated immunosuppression, highlighting the Xenomimetic mouse (X-mouse) model and the Omental Tumor Xenograft (OTX) model that were featured in a recent study to evaluate the efficacy of a novel phosphatidylserine-binding molecule for targeting immunosuppressive tumor-associated exosomes.

## 1. Introduction

Extracellular vesicles (EV) released from normal and neoplastic cells called immunoglobulin secretory vesicles (ISV) were first identified and characterized biochemically as well as ultra-structurally in 1987 [1]. A subset of EV surrounded by a lipid bilayer and released by most eukaryotic cells called exosomes was originally discovered and at first, simply considered as cellular waste products [2]. However, extensive research over the past two decades supports the growing awareness that exosomes contribute to a wide range of biological processes in health and disease including cancer [3]. Exosomes and their cargos such as lipids, metabolites, proteins and nucleic acids represent prognostic markers as well as potential therapeutic targets [4,5,6,7]. In this review, we focus upon the role of a subset of exosomes present in tumor microenvironments (TME) that are immune suppressive, therefore representing a potential therapeutic target. While considerable effort has been made to link exosome cargos to their biological function, most studies were in vitro and focused upon proteins and nucleic acids. In the first part of this review, we summarize these in vitro data, while also emphasizing the role of two exosomal lipids, phosphatidylserine (PS) and ganglioside GD3, that were causally linked with exosome-mediated suppression of human T cell function [8,9]. In this section, we also shed light on some recent results supporting the viability of targeting PS+ immunosuppressive exosomes in human TME using a novel PS-binding molecule called ExoBlock [10].

In the latter part of our review, we discuss in vivo studies and model systems that have contributed to our understanding of exosomal immunosuppression. In this context, we focus upon the design of two novel human tumor xenograft platforms that has now made it possible to evaluate the pre-clinical efficacies of multiple different immune-based therapies [11,12] and to begin to monitor the presence and function of tumor-associated exosomes in vivo. In this segment, we also discuss the in vivo studies demonstrating the efficacy of blocking exosomes with ExoBlock using these two tumor xenograft models [10].

## 2. Exosome Biogenesis, Isolation and Characterization:

Exosomes originate from the endocytic pathway, which distinguishes them from other secreted EVs such as microvesicles and apoptotic bodies. The biogenesis of exosomes starts with the invagination of the plasma membrane, a process that is facilitated by neutral sphingomyelinase 2 (nSMase2) [13]. This leads to the formation of intracellular multivesicular bodies (MVB) containing intraluminal vesicles (ILVs) [3]. The formation of MVBs and ILVs is tightly regulated by the endosomal sorting complex required for transport (ESCRT), an intricate protein machinery that includes ESCRT-0, ESCRT-I, ESCRT-II and ESCRT-III, and is required for MVB formation, protein cargo sorting, and vesicle budding [14,15,16]. Additional molecules such as ALIX (Apoptosis-linked gene 2-interacting protein X), VTA1 (Vesicle Trafficking 1), VPS4 (Vacuolar protein sorting-associated protein 4), and TSG101 (Tumor susceptibility gene 101 protein) are also closely associated with ESCRTs and are involved in the regulation of exosome biogenesis [17,18,19]. MVBs to be exocytosed are then transported to the plasma membrane. Late endosomal transport is regulated by multiple proteins, and MVB docking was shown to be mediated by RAB27 in multiple cancers [20]. Following docking, MVBs couple to the SNARE (soluble N-ethylmaleimide-sensitive component attachment protein receptor) proteins, which include SNAP23, VAMP7, YKT6, Syntaxin-1a, Syntaxin-4 and Syntaxin-5 [20]. The MVBs fuse with the plasma membrane and the vesicles are ultimately secreted via exocytosis as exosomes ranging from 40–160 nm in diameter.

Exosomes can be isolated using a variety of techniques that are based on utilizing one or more of their characteristic features such as density, shape, size, and surface molecules. The most popular method of isolation based on published literature is ultracentrifugation (UC) [8,9,10,21,22]. UC protocols for exosome isolation often include prior low- and medium-speed centrifugation steps to exclude cells and cell debris respectively, as well as membrane filtration to exclude larger vesicles and organelles [8,9,10,21,23]. Another popular technique used to isolate exosomes is called size exclusion chromatography (SEC) [10,22], which is a size-based isolation technique. When size exclusion columns are used, different fractions are analyzed for their protein content as well as vesicle number, and the fractions with the highest vesicle to protein ratio are pooled to constitute the final exosome preparation [10]. It is not uncommon to use a combination of UC and SEC for exosome isolation [24]. Other size-based isolation techniques include ultrafiltration using membrane filters and the less commonly used flow field-flow fractionation [22]. Exosomes can also be isolated by precipitation, either using water-excluding polymers such as polyethylene glycol (PEG) [25] or by a pH-dependent “salting out” process with sodium acetate [26]. Immune-affinity-based capture techniques can be used to isolate exosomes based on the presence of surface molecules of interest. Microfluidics-based techniques were also used successfully for exosome isolation. A detailed overview and comparison of these techniques can be found in Li et al. [22].

As mentioned before, the endocytic pathway distinguishes exosomes from other secreted EVs, as this mechanism results in the acquisition of unique markers associated with the endocytic pathway. The International Society of Extracellular Vesicles (ISEV) lays out guidelines for the characterization of EVs [27]. Typically, exosome preparations are characterized based on a combination of these parameters: morphology, size, presence of a lipid bilayer membrane and biochemical composition (Figure 1). Morphology studies usually involve transmission electron microscopy (TEM) [21], scanning electron microscopy (SEM) [9], cryo-electron microscopy [28] and atomic force microscopy [28]. TEM is sometimes used in conjunction with immunogold-labeling to identify exosome-associated molecules [28]. Exosomes range in size from 40–160 nm, and size distribution may be measured by dynamic light scatter (DLS) [29], or more commonly, nanoparticle tracking analysis (NTA) [9,21]. The presence of a lipid bilayer can be established using anisotropy measurements [21], and specific lipids can be identified using flow exometry of exosomes attached to latex beads [9,21]. Biochemical characterization of exosomes generally involves establishing the presence of exosome-associated molecules such as tetraspanins (CD63, CD81 and CD9), ALIX, TSG101, Flotillin, and tumor histotype-specific markers (e.g., EpCAM for ovarian carcinomas) as well as the absence of cellular contaminants such as Calnexin, GM130 or Cytochrome C, which may be achieved by immunoblotting and/or flow exometry [9,21,30,31]. The characterization of exosomes is usually followed by functional studies to determine their effect on the cells of interest.

## 3. In Vitro Studies with Human Tumor-Associated Exosomes

The human TME harbors different cellular compartments such as tumor cells, immune cells and stromal cells in addition to various acellular factors such as prostaglandins, adenosine, cytokines and exosomes, the dynamic interplay between all of which governs tumor growth [32,33]. Some of these factors, including exosomes, were shown to induce immunosuppressive conditions in the TME, facilitating tumor progression. Exosomes in the TME are quite heterogeneous in terms of their origin, as they are released from tumor as well as non-tumor cells. A phenotypic analysis of exosomes derived from ovarian tumor ascites fluid revealed the presence of exosomes originating from different cells such as erythrocytes, fibroblasts, platelets and leukocytes in addition to those derived from tumor cells [8]. The exocytosis of immunosuppressive exosomes is one of the mechanisms by which tumors evade anti-tumor immune responses in the TME. Exosomes are also heterogeneous in terms of their constituents, since different proteins, nucleic acids and lipids may be present on their surface or as intra-vesicular cargo. According to ExoCarta, a database of exosome-associated molecules, 9769 proteins, 3408 mRNAs, 2838 miRNAs, and 1116 lipids were shown to be associated with exosomes (available at http://www.exocarta.org; accessed on 12 November 2021). Some of these molecules were identified as major players in exosome-mediated modulation of the TME (Figure 2).

Tumor-associated exosomes play a crucial role in facilitating tumor growth by affecting different immune regulatory mechanisms such as immune activation [8,9,21,34], antigen expression [35], immune surveillance [36], immunosuppression [37] and communication between immune cells [38]. These exosomes can exert immunosuppressive effects on multiple immune compartments in the TME. Exosomes in the TME can directly suppress T cell activation [8] or they can drive differentiation of monocytes towards myeloid-derived suppressor cells (MDSCs) [39] and stimulate regulatory T cells (Tregs) [40], which in turn can suppress T cell activation [8]. Moreover, tumor-derived exosomes can also promote M2-like macrophage polarization that can promote tumor progression [41,42]. In this section of the review, we will focus on the effect of tumor-derived exosomes on T cell function. Although the immunosuppressive mechanisms associated with exosomes have not been fully decoded yet, multiple studies over the last decade have provided valuable insights into the effects of tumor-associated exosomes on T cell function in human TME.

Tumor-associated exosomes from different cancer types were shown to modulate T cell function [10,43,44] and were shown to suppress CD8 as well as CD4 T cells derived from normal donor PBLs [8,9,21]. The exosomes were found to inhibit early (nuclear translocation of NFκB and NFAT), intermediate (degranulation, upregulation of activation markers CD25 and CD69, intracellular expression of the cytokines IL-2 and IFN-γ) as well as late (proliferation) endpoints of T cell function when activated through the T cell receptor (TCR) [8,10,21]. One of the main characteristics of exosome-mediated suppression of T cell function was that it was specific to activation through the TCR. Exosomes inhibited the polyclonal (using immobilized antibodies to CD3 and CD28) as well as antigen-specific activation of T cells through the TCR but had no demonstrable effect on the activation of T cells stimulated using PMA and Ionomycin, which bypasses the TCR [8,21]. This signature TCR signaling arrest was also reported in exosomes derived from chronic inflammatory microenvironments [23]. Exosome-mediated inhibition was found to correlate with the binding and internalization of the exosomes by T cells since the T cells which bound and internalized exosomes showed poor to no activation [21]. Interestingly, this inhibition was found to be transient and could be reversed by the removal of exosomes [21]. Exosomes derived from human [43] and mouse (unpublished data) melanoma cell lines were also found to inhibit T cell function in vitro.

Tumor-derived exosomes not only inhibit anti-tumor T cell function but also enhance and support the function of Tregs in vitro. Human tumor-derived exosomes were shown to inhibit IL-2 mediated proliferation of CD4 and CD8 T cells. Further phenotypic analysis revealed that in the presence of tumor-derived exosomes, it was the Treg population that responded to IL-2 in a TGF-β dependent manner [45]. Furthermore, T cells when incubated with human mutant KRAS tumor-derived exosomes induced increased secretion of IL-10 from naïve CD4 T cells and increased the switch of naïve CD4 T cells into FoxP3^+^ phenotype, both of which lead to an immunosuppressive TME [46]. Likewise, co-incubation of ovarian tumor-derived exosomes but not those derived from normal cells resulted in the expansion of human Tregs which mediated conversion of helper CD4^+^CD25^neg^ T cells into immunosuppressive CD4^+^CD25^high^FOXP3^+^ Tregs. These exosomes also induced the upregulation of FasL, IL-10, TGF-β1, CTLA-4, granzyme B and perforin on the surface of Tregs, and conferred them resistant to apoptosis in comparison to conventional T cells [47]. The immunosuppressive effect of exosomes on immune cells involved in anti-tumor responses, such as antigen-presenting cells, T cells and NK cells; along with their effects on angiogenesis, stromal reprogramming and induction of suppressive cells such as Tregs and myeloid-derived suppressor cells (MDSCs) contribute to the formation of pre-metastatic niches, favoring metastasis and tumor spread [48,49].

While our understanding of how these exosomes interact with and regulate T cells is not yet complete, in vitro studies have revealed several types of interactions between exosomes and T cells: (1) binding of exosomes on the surface of T cells through adhesion molecules, (2) fusion of exosomes with the plasma membrane, and (3) internalization of exosomes through receptor-mediated endocytosis [21,43,50]. These interactions could facilitate surface signaling as well as the transfer of exosomal contents into the target cells which can mediate their communication with the target cells and aid in immunosuppression. We will describe below how different exosomal contents exert an immunosuppressive effect.

### 3.1. Immunosuppressive Proteins Associated with Tumor Exosomes

Exosomes consist of cytosolic and membrane proteins derived from the parent cells. They are largely enriched in certain proteins such as adhesion molecules (some tetraspanins and integrins), membrane trafficking molecules (annexins and Rab proteins), chaperones (Hsp70 and Hsp90), signal transduction molecules (G heterodimers, protein kinases) and enzymes (GAPDH, enolase, kinases and phospholipases) [51]. Tumor-derived exosomes consist of tumor antigens as well as proteins involved in immunosuppression such as death receptor ligand FasL or TRAIL, checkpoint receptor ligands such PD-L1, and adenosine ectoenzymes such as CD39 and CD73 (Figure 2) [52,53].

Tumor-derived exosomes from melanoma, pancreatic cancer, colorectal cancer, and kidney adenocarcinoma tumor cells as well as sera of oral cancer patients were shown to induce T cell apoptosis [54,55,56,57,58]. The T cells were activated in presence of oral cancer patient sera-derived exosomes for 1–4 days in vitro and apoptosis was measured using DNA fragmentation assay. These exosomes were positive for FasL and triggered CD8 T cell apoptosis, which was partially inhibited in the presence of Z-VAD-FMK (pan caspase inhibitor) as well as by using anti-FasL antibodies [59]. A recent study demonstrated that exosomes derived from pancreatic cell lines can activate p38 MAP kinase in T cells, which in turn induces endoplasmic reticulum (ER) stress-mediated apoptosis in T cells [57]. Furthermore, exosomes from nasopharyngeal carcinoma (NPC) patient-derived sera and NPC cells express galectin-9, which is a ligand of the TIM-3 receptor. These exosomes were shown to induce apoptosis in mature Th1 lymphocytes and CD4^+^ T cells in vitro [60]. Similar apoptotic effects on human T lymphocytes were described for exosomes expressing TRAIL [35,61].

The checkpoint molecule ligand PD-L1 interacts with its receptor PD-1 on T cells and notably decreases T cell proliferation and maturation while inducing apoptosis [62]. Tumors cells utilize the PD-L1/PD-1 axis to evade anti-tumor T cell responses by secreting exosomes loaded with PD-L1 [43]. PD-L1+ exosomes from different tumor types such as melanoma, prostate, lung, head and neck, oral-esophageal, and gastric cancers inhibit anti-tumor T cell responses, which aids tumor progression. Chen et al. demonstrated that both human and mouse melanoma cell lines as well as melanoma patient sera are enriched in exosomal PD-L1. They also demonstrated that exosomal PD-L1 inhibited CD8 T cell proliferation and cytokine secretion, which could be rescued by using anti-PD-L1 antibodies [43]. Similarly, breast and lung cancer cell lines also release PD-L1+ exosomes that inhibit CD8 proliferation and function, which can be similarly rescued using a function-blocking anti-PD-L1 antibody [43]. These in vitro studies strengthen the notion that tumors escape the anti-tumor T cell responses by secreting exosomal PD-L1 in the TME.

Exosomes derived from different cancers were found to express CD39 and CD73, which are enzymes that catalyze the hydrolysis of ATP to generate the immunosuppressive molecule adenosine [63]. Adenosine suppresses T cell function in an Adenosine A_2A_ receptor-dependent manner, via dephosphorylation of STAT5. Additionally, adenosine can promote Treg differentiation, inhibit the activation of T cells stimulated through IL-2 receptor or TCR, inhibit NK cell cytokine production and cytotoxicity, and impair the maturation and function of DCs [63].

### 3.2. Immunosuppressive Nucleic Acids Associated with Tumor Exosomes

The nucleic acids in the exosomal cargo consist of DNA and RNA, which can modulate T cell immune function. Tumor-derived exosomes consist of genomic DNA as well as mitochondrial DNA, complementary DNA and transposonal DNA (Figure 2). They also carry relevant clinical information about tumor-specific mutations in multiple genes such as epidermal growth factor receptor (EGFR), proto-oncogene B-Raf, RAS, isocitrate dehydrogenase 1 and human epidermal growth factor receptor 2 (HER2) [64]. The different RNAs that were identified in tumor-associated exosomes include messenger RNAs (mRNAs), microRNAs (miRNAs), and long non-coding RNAs (lncRNAs) [65]. Recent studies have shown that tumor-associated exosomal miRNAs and lncRNAs have the potential to regulate immune responses by modulating gene expression and signaling pathways via transfer of RNA transfer. The lncRNA PCED1B-AS1, associated with exosomes derived from hepatocellular carcinomas, was shown to enhance PD-L1 expression in cancer cells while inhibiting T cell and macrophage function [66], while lncRNA SNHG16, associated with breast cancer-derived exosomes induces immunosuppressive CD73^+^ γδ1 Treg cells [67]. Exosomes derived from NPC cell lines as well as patient sera were found to be enriched in miR-24–3p, and the levels of miR-24-3p correlated with the lowest disease-free survival rate. These exosomes inhibited T cell proliferation as well as Th1 and Th17 differentiation while driving Treg generation. Mechanistic analysis revealed that miR-24-3p enriched exosomes increased p-ERK, p-STAT1 and p-STAT3 expression while decreasing the expression of p-STAT5 during T cell proliferation and differentiation. Moreover, in vitro studies determined that exosomal miR-24-3p directly targets fibroblast growth factor 11 (FGF11), which impedes T cell proliferation and Th1 and Th17 differentiation while inducing Treg differentiation [68]. Dou et al. demonstrated that miR-92 levels were highly elevated in exosomes derived from cancer-associated fibroblasts [CAFs] in breast cancer, which demonstrated an ability to suppress T cell proliferation and induce apoptosis [69]. Exosomes derived from ovarian cancer-associated macrophages are enriched in miR-29a-3p and miR-21-5p, which suppress STAT3-mediated signaling in T lymphocytes [70]. Murine melanoma-derived exosomes induce apoptosis through the mitochondrial pathway in CD4+ T cells by transferring miR-690 [71]. Cervical cancer-derived exosomal miR-1468-5p was found to promote the upregulation of PD-L1 in the lymphatics which in turn suppress T cell immunity [72]. Other miRNAs associated with NPCs such as miR-891a, miR-106a-5p, miR-20a-5p and miR-1908 were shown to down-regulate the microtubule affinity regulating kinase 1 (MARK1) signaling pathway in T lymphocytes. These studies implicate exosomal nucleic acids in T cell immunosuppression in the TME.

### 3.3. Immunosuppressive Lipids Associated with Tumor Exosomes

Exosomes consist of a variety of lipids such as cholesterol, sphingolipids, phosphatidylcholine (PC) and PS on the outer leaflet of exosomes while other lipids are present in the inner leaflet [52,73]. Recent studies have shown that exosomes transfer cholesterol, fatty acids and eicosanoids from parent cells to the recipient cells, thereby leading to inflammation and causing metabolic and immune changes in certain microenvironments [74,75].

The phospholipid PS is normally expressed in the inner leaflet of normal non-apoptotic cell membranes. However, PS is expressed on the surface of apoptotic cells as well as non-apoptotic cancer cells such as malignant melanoma, leukemia, neuroblastoma and gastric carcinoma [76]. The immunosuppressive effects of PS are well documented [73] and are being utilized to suppress unwanted immune responses in several diseases [77,78,79]. PS binds to a variety of different receptors on immune cells [73]. Studies have shown that PS exposed on the surface of tumor cells can mediate T cell immunosuppression which leads to tumor progression [73]. Exosomes derived from a variety of human cancers including ovarian carcinomas and melanomas were shown to express PS abundantly on their surface [8,10], and PS+ exosomes were recently demonstrated to be highly enriched in TME [10]. To demonstrate whether PS can directly inhibit T cell immunosuppression in vitro, PS/PC liposomes were formulated (in a 30:70 ratio) and tested for their ability to inhibit T cell activation. PS/PC liposomes were found to significantly inhibit T cell activation in a dose-dependent manner, whereas control (100% PC) liposomes had no such inhibitory effect [8]. While the mechanisms of PS-mediated suppression of signaling downstream of the TCR are not fully understood, one hypothesis is that it acts through the modulation of the enzyme diacylglycerol kinase (DGK). DGK converts diacylglycerol (DAG), a signal-transducing secondary messenger downstream of TCR engagement, to phosphatidic acid, a signal dampener, thus playing a critical role in determining whether activation or anergy ensues following TCR stimulation. PS was shown to enhance the metabolic activity of DGK [80]. Consistent with this hypothesis, inhibition of DGK using pharmacological inhibitors was found to rescue exosome-mediated immunosuppression of T cells [8].

Blockade of PS using an anti-PS antibody was found to reverse the exosome-mediated suppression of NFκB translocation as well as IFN-γ expression in T cells activated through the TCR, demonstrating proof of principle [8]. Furthermore, depletion of PS+ exosomes by magnetic separation using anti-PS antibody conjugated to magnetic beads was found to substantially diminish exosome-mediated immunosuppression, establishing that this suppression was in part PS-dependent [8]. Clinical trials that targeted PS in cancers using the anti-PS antibody, Bavituximab met with modest success [73], perhaps due to relatively low binding affinity or high molecular weight of the antibody, which affects its distribution and half-life. Recently, a novel molecule, ExoBlock (patent pending) was developed to target PS in TME [10]. ExoBlock, a hexamer with six binding sites for PS, (which is more than anti-PS antibody) was demonstrated to bind PS with high avidity [10]. The efficacy of ExoBlock in terms of rescuing exosome-mediated suppression of T cell function was tested in vitro. ExoBlock was able to rescue multiple early and late endpoints of T cell activation (NFκB translocation, CD69 and CD25 upregulation, IL-2 and IFN-γ expression, and proliferation) following short-term or long-term stimulation [10]. These results indicate that exosomal PS is a potential therapeutic target, and blocking it can rescue T cell function.

Sphingolipids were also shown to be elevated in the plasma and ascites fluid of ovarian cancer patients. Gangliosides in particular are widely expressed in many cancer tissues including melanoma and are reported to be involved in cancer progression [81]. Several gangliosides have immunosuppressive properties. GD3, a disialylated ganglioside, was found to be abundantly expressed in ovarian cancers ascites fluids as well as in exosomes derived from them [9] and was also shown to be expressed in murine and human melanoma-derived exosomes (unpublished data). GD3 can bind to multiple receptors on different immune cells, including Siglecs, which play an important role in immune regulation [82]. A causal link between ganglioside GD3 and exosome-mediated immunosuppression was established by demonstrating the ability of GD3:PC (30:70) liposomes to inhibit T cell function [9]. Blocking exosomal GD3 using an antibody (anti-GD3), or depleting GD3+ exosomes by magnetic separation using anti-GD3 bound to magnetic beads was found to rescue T cells from exosome-mediated immunosuppression. The inhibitory activity of vesicular GD3 was found to be dependent on sialic acid residues, as enzymatic removal of these residues on GD3+ exosomes as well as GD3/PC liposomes resulted in a substantial reversal of immunosuppression [9]. These results establish exosomal GD3 as a potential therapeutic target.

Exosome-associated immunosuppressive effects on T cells could also be indirectly mediated by their effect on dendritic cells. A study by Salimu et al. showed that exosomes from prostate cancer cell lines impaired DC function, and triggered CD73 and CD39 expression on DCs, which promoted adenosine-mediated suppression of CD8 T cell activation [53]. They established that it was the exosomal lipid prostaglandin E2 (PGE2), which was the potential driver of CD73 induction on the DCs. These results suggest that the dominant effect of exosomes is immunosuppression and not antigen delivery, revealing potential crosstalk between exosomes and different immune cells [53].

## 4. In Vivo Models for the Study of Tumor-Associated Exosomes

While most of our understanding of the mechanisms by which exosomes mediate immunosuppression in human TME is derived from in vitro and ex vivo studies such as the ones listed above, there is a relatively low volume of in vivo studies. This can at least partly be attributed to the limited number of model systems, which for the most part are restricted to xenograft models, as well as the challenges associated with in vivo tracking of exosomes. Nevertheless, these studies have taught us valuable lessons and provided useful insights regarding the effects of exosomes in immune regulation at local as well as systemic levels (Table 1).

Multiple studies have employed orthotopic and heterotopic tumor models to study the biodistribution of exosomes [83,84] as well as to dissect the role of exosomes in tumorigenesis [43], metastasis [85], and immunosuppression [86]. These models are sometimes used in conjunction with intravenous/systemic delivery of exosomes, isolated either from cultured cell lines or from patient samples, to study their short-term and long-term accumulation as well as effects on the tumor microenvironment. The exosomes are often labeled by one or more approaches which have been reviewed in detail [83].

### 4.1. In Vivo Studies with Murine Exosomes

A number of studies utilizing labeled tumor-derived exosomes report their localization to different organs, including secondary lymphoid organs such as spleen [87,88,89,90,91,92,93] and lymph nodes [94,95]. Exosomes were demonstrated to be taken up by both immune and non-immune cells in vivo [96]. Some of these studies have identified one or more molecules associated with exosome-mediated immunosuppression. (Table 1). For instance, ovarian cancer-derived arginase-1 (ARG1)-positive exosomes were found to inhibit the activation and proliferation of adoptively transferred T cells in response to antigen-specific stimulation in lymph nodes, an effect that could be reversed by pharmacological inhibitors of ARG1 [97].

Additionally, Zhou et al. reported that B16 melanoma-derived exosomes cause apoptosis of CD4 T cells in vivo, and the resultant reduction in the numbers of these cells promotes tumor growth using an orthotopic B16 melanoma mouse model [71]. This effect was found to be linked to the microRNA cargo of these exosomes.

Chen and colleagues examined the in vivo effects of exosomal PD-L1 by establishing a syngeneic mouse melanoma model in C57BL/6 mice using B16-F10 cells in which PD-L1 was stably knocked down using lentiviral shRNA. Intravenous injection of exosomes derived from wild-type B16-F10 cells accelerated the growth of these tumors, while pre-treatment of these exosomes with anti-PD-L1 antibodies reversed this effect. Additionally, the number of tumor-infiltrating lymphocytes (TILs) as well as proliferating CD8+ T cells in the spleen and lymph nodes, which decrease significantly after the injection of exosomes, were also rescued by anti-PD-L1 pretreatment of the exosomes, establishing a causal link between systemic suppression of anti-tumor immunity and exosomal PD-L1 [43].

Orthotopic syngeneic models were also adopted by Yang and colleagues to study the role of exosomal PD-L1 in breast cancer [98]. Tumors were established by injecting the murine breast cancer cell line 4T1 into the mammary fat pads of BALB/c mice. Inhibition of exosome secretion, either by using the pharmacological inhibitor GW4869, or by using a genetic approach by knocking down Rab27a, was found to inhibit tumor growth and augment the efficacy of anti-PD-L1 therapy. Tumor growth was found to be severely suppressed by knocking out PD-L1 in these cells but was rescued by co-injection with PD-L1+ exosomes. Exosomal PD-L1 was also found to decrease cytotoxic T cell activity in tumors in this study [98].

A syngeneic model of metastatic breast cancers was established by Wen et al. [92] to investigate the biodistribution as well as pro-metastatic and immunosuppressive roles of tumor-derived exosomes. The authors found a significant uptake of these exosomes by CD4 and CD8 T cells in the lungs of experimental mice, although the uptake by phagocytic cells such as DCs and macrophages was substantially higher as expected [92]. The authors also found that acute intravenous injection of exosomes resulted in a substantial decrease in CD4 and CD8 T cell frequencies in the lung. These and other exosome-induced changes in the lung were found to play a crucial role in promoting metastasis of these tumors to this site.

Poggio and colleagues described a syngeneic model of prostate cancer established by subcutaneous engraftment of the murine prostate cancer cell line TRAMP-C2 [44]. These studies extended our understanding of the role of immune-suppressive exosomes, as the inhibition of exosome biogenesis by deleting Rab27a or nSMase2 in the tumor cells was found to suppress tumor growth and extend lifespan. There was also a concomitant increase in spleen size as well as in the numbers, cytotoxic activity (represented by Granzyme B) and proliferation of CD8+ cells, along with a decrease in the expression of the exhaustion-associated checkpoint molecule Tim-3. Exogenously introduced PD-L1+ exosomes in vivo reversed these effects. Similar results in terms of the inhibition of tumor growth in vivo were seen with syngeneic models established with the murine colorectal cancer cell line MC38 [44].

Kim and colleagues tested the effect of immune-suppressive exosomes on the growth of non-small cell lung cancer (NSCLC) cells using a syngeneic model established by the subcutaneous engraftment of murine LLC-1 cells in C57BL/6 mice [99]. The authors found that systemic administration of exosomes derived from LLC-1 cells genetically modified to overexpress PD-L1 accelerated tumor growth considerably, and was accompanied by a decrease in TILs.

### 4.2. In Vivo Studies with Human Exosomes in Murine Models

Some authors have studied human tumor-derived exosomes by injecting them into naïve or tumor-bearing mice [86,93,96,100]. Wiklander et al. reported that the biodistribution as well as homing characteristics of xenotransplanted tumor-derived exosomes from different species, including human, was found to be similar [93], providing some degree of credulence to these studies. In one such study by Azambuja and colleagues, exosomes were derived from human glioblastoma cell lines and injected i.v. into healthy mice [86]. The introduction of these exosomes that were enriched in immune suppressive proteins such as CD39, CD73, FasL, CTLA-4 and TRAIL, was found to result in a decrease in splenic CD8+ T cells, while CD4+ T cells and regulatory T cells (Tregs) were unaffected, suggesting that exosomal inhibition of T cells was not limited to TILs.

Chen and colleagues also established orthotopic melanoma xenografts in nude mice using the human metastatic melanoma cell line WM9 to study the expression and function of PD-L1+ exosomes in melanoma TME [43]. PD-L1+ exosomes were detected in the circulation of these xenograft-bearing mice, and the level of circulating exosomal PD-L1 was found to positively correlate with tumor size [43].

While these studies suggested a possible inhibitory effect of human tumor-associated exosomes on T cells, there were no models that made it possible to determine the effect of these exosomes on the function of tumor-specific T cells in solid TME. We recently described two patient-derived xenograft models—the X-mouse model [10,12] and the OTX model [10,11,101]—that allowed us to study the effect of blocking or targeting exosomes in the tumor microenvironment on tumor-specific T cells.

#### 4.2.1. Xenomimetic Mouse (X-mouse) Model

Recently, we reported a novel xenograft model, called the Xenomimetic (X-) mouse model that was designed and validated specifically to enable the evaluation of therapies designed to enhance anti-tumor T cell responses [12]. This model is established using a human melanoma tumor cell line that is genetically modified to express patient melanoma-derived neoantigen peptides in the context of HLA- A*02:01 (DM6-Mut cells). In traperitoneal injection of DM6-Mut cells into globally immunodeficient NSG mice (Figure 3) results in rapid engraftment of these cells in the greater omentum, an organ that was shown to support the growth of multiple human tumors [10,11,12,102]. While untreated tumors grow rapidly resulting in metastasis and accumulation of ascites fluid in these mice, the introduction of patient-derived neoantigen-specific T cells that are specific for the peptides expressed on the tumor targets results in a suppression of tumor growth. However, this initial suppression is followed by tumor escape, which correlates with the upregulation of exhaustion-associated checkpoint molecules LAG-3 and PD-1 on the adoptively transferred T cells, and a concurrent expression of PD-L1 in the tumor microenvironment. This model allows one to evaluate the therapeutic efficacy of T cell function-enhancing immune-based therapies based on their ability to suppress this tumor escape. The X-mouse model was originally validated by demonstrating the efficacy of two established T cell function-enhancing immune-based therapies. The first one was checkpoint blockade using anti-PD1 (Nivolumab), FDA-approved immune-based therapy for metastatic melanoma. Treating X-mice with Nivolumab was found to significantly suppress tumor escape, credentialing our model [12]. The second immune-based therapy tested in this model was IL-12, delivered liposomally, to activate T cells. Not only did IL-12 treatment suppress tumor escape in the X-mouse model, but it also suppressed the upregulation of LAG-3 and PD-1 on T cells [12].

More recently, this model was used to evaluate the efficacy of ExoBlock (discussed above), a novel multimeric PS-binding molecule used to target immune-suppressive exosomes in the tumor microenvironment [10]. DM6-Mut tumor cells were found to release immunosuppressive PS+ exosomes, and administration of ExoBlock was found to suppress tumor escape with an efficacy comparable to Nivolumab in the X-mouse model. Since the adoptively transferred patient-derived T cells are the only immune cells present in this model, the suppression of tumor rebound can be interpreted as a direct effect of the therapy on T cells [10]. This was confirmed by demonstrating that the efficacy of ExoBlock is dependent on the presence of T cells, since ExoBlock had no effect on DM6-Mut tumors if T cells were not adoptively transferred into this model [10].

While the X-mouse model does not represent a complete tumor microenvironment due to the lack of human stromal cells as well as innate immune cells (which play important roles in tumor progression and response to immune-based therapies), it does have several advantages including but not limited to providing a controlled tumor microenvironment where the number of tumor cells, as well as the number and tumor-specificity of T cells, is well defined, and the ability to rapidly and sensitively quantify tumor burden. The OTX model discussed next provides a platform that includes a more complete tumor microenvironment.

#### 4.2.2. Omental Tumor Xenograft (OTX) Model

The omental tumor xenograft (OTX) model is a patient-derived xenograft (PDX) model established by the injection of tumor aggregates derived from ovarian cancer patients into the peritoneal cavity of NSG mice [11,101]. This results in rapid engraftment of the tumor as well as the patients’ tumor-associated stroma into an anatomically well-defined site, the greater omentum, establishing a more complete tumor microenvironment (Figure 4). Tumor-infiltrating lymphocytes (TILs) from the original tumor are present in these xenografts and were shown to be viable as well as responsive to IL12-mediated activation [101]. Exogenously transferred fully functional T cells were shown to become anergic and hyporesponsive to activation upon entering the tumor microenvironment in this model [11]. Immunosuppressive PS+ exosomes were isolated from ascites fluids that develop in these xenograft-bearing mice (unpublished data). We recently tested the efficacy of ExoBlock, the aforementioned PS-binding hexamer in the OTX model [10]. The study revealed that blocking immunosuppressive exosomes with ExoBlock results in a substantial decrease in the tumor burden and metastasis, accompanied by an increase in CD4 and CD8 T cells as well as IFN-γ, an indicator of T cell activation, in the TME. These results are consistent with a recent study that demonstrates that tumor antigen-specific T cell priming is enhanced by targeting PS in a murine melanoma model [103]. Since the only T cells present in these xenografts are TILs from the original tumors, this study provides evidence that targeting immunosuppressive exosomes can restore the function of at least a subset of TILs, which bodes well for immune-based therapies designed to enhance T cell function. These results were found to be reproducible across xenografts established using tumors from several different ovarian cancer patients. The presence of a complete tumor microenvironment in this model also allowed us to investigate the effect of ExoBlock treatment on other components such as macrophages. PS was shown to skew macrophage polarization towards an immunosuppressive M2 phenotype [10,73], and blockade of PS with ExoBlock was able to reverse this, increasing the M1:M2 ratio in the tumor microenvironment [10]. ExoBlock treatment also significantly reduced the number of circulating PS+ exosomes in xenograft-bearing mice [10], presumably the result of PS blockade as well as the suppression of tumor growth, resulting in a lower tumor burden in these mice.

The X-mouse model and the OTX model offer exciting opportunities for investigators to evaluate T cell function-enhancing immunotherapies, such as direct stimulation (e.g., IL-12), blockade of immune checkpoint molecules (e.g., PD-1, PD-L1) or blockade of immunosuppressive exosomes (ExoBlock) as monotherapies as well as combination therapies to determine synergy.

## 5. Concluding Remarks

Immunotherapy in general, and specifically immune checkpoint inhibitors such as antibodies to PD-1 and PD-L1, were found to be therapeutically effective in several different types of cancer including melanoma, non-small cell lung cancer and renal cancer. However, durable clinical responses have only been seen in 10–30% of these patients [104]. In other cancer patients, such as those with prostate cancer, clinical responses have been rare [105,106]. As mentioned in this review, immune-suppressive exosomes are present in the microenvironment of many different types of human tumors. However, no drugs currently exist that are designed to target and inhibit tumor-associated exosomes in vivo. Tumor-associated exosomes could possibly be targeted as a standalone or combination therapy with blockade of immunosuppressive surface proteins such as PD-L1, or lipids such as PS and GD3, representing potential therapeutic targets. Cancer immunotherapy could possibly benefit from the development of alternative strategies such as these that complement anti-checkpoint therapy in order to enhance anti-tumor T cell immune responses in the TME and achieve cancer remission in the clinic.

## Figures and Tables

**Figure 1 cells-10-03155-f001:**
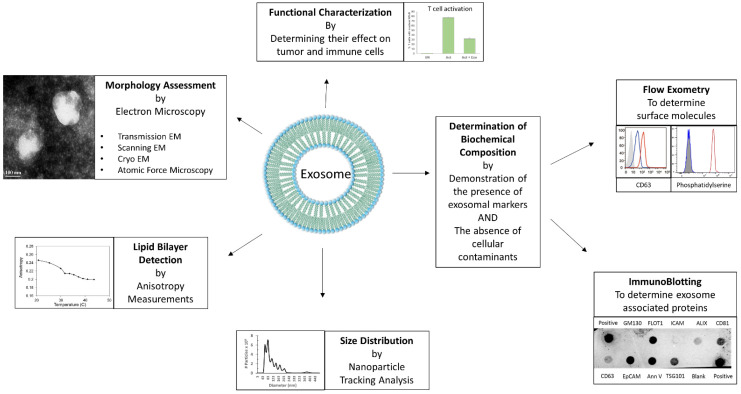
Characterization of Exosomes. The different parameters and methods utilized for characterizing exosomes, such as morphology assessment by electron microscopy, lipid bilayer detection by anisotropy measurement, size distribution by nanoparticle tracking analysis (NTA), determination of biochemical composition by flow exometry and/or immunoblotting, are depicted. Representative data shown are either reused from previously published article by the authors [8], or unpublished original data.

**Figure 2 cells-10-03155-f002:**
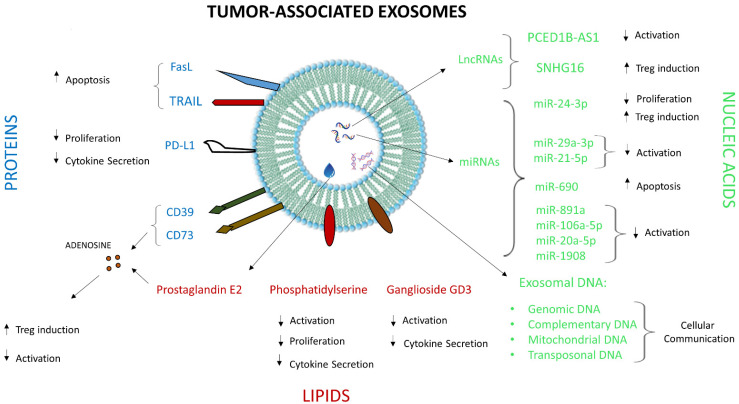
Immunosuppressive molecules associated with tumor exosomes and their effects on human T cells. Different proteins (blue text), lipids (red text) and nucleic acids (green text) and their effects on T cell activation, proliferation, cytokine secretion and other functions are depicted.

**Figure 3 cells-10-03155-f003:**
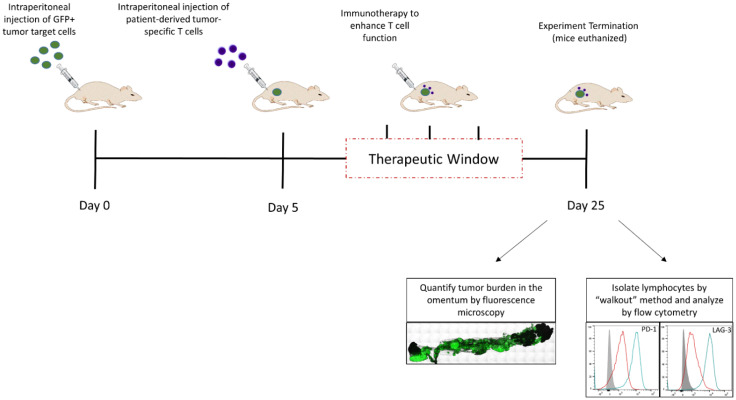
The Xenomimetic Mouse (X-mouse) Model. Xenografts are established in the greater omentum of NSG mice by intraperitoneal injections of GFP+ tumor targets on day 0 and patient-derived neoantigen-specific cells on day 5. Immunotherapeutics designed to enhance T cell function are administered in the “therapeutic window” between day 10 and day 25. On day 25, the mice are euthanized and omenta removed to allow quantification of tumor burden. Lymphocytes are isolated using the “walkout” method and phenotyped by flow cytometry. Representative data shown are original unpublished data generated by the authors.

**Figure 4 cells-10-03155-f004:**
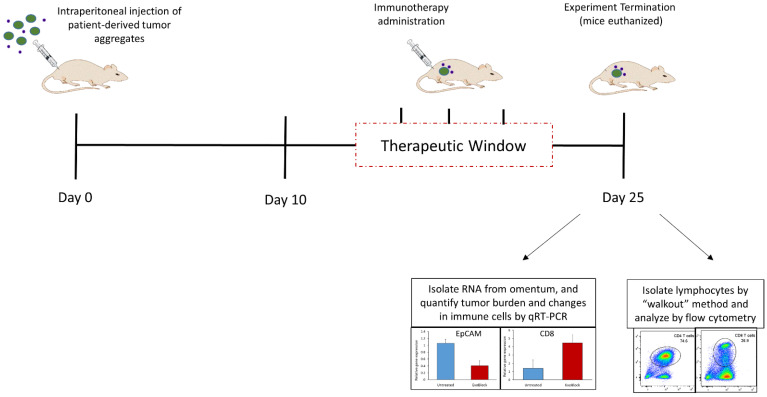
The Omental Tumor Xenograft (OTX) Model. Xenografts are established in the greater omentum of NSG mice by intraperitoneal injections of patient-derived tumor aggregates which include immune and stromal cells in addition to the tumor cells. Tumors are allowed to engraft and establish for 10 days. Immunotherapeutics are administered in the “therapeutic window” between day 10 and day 25. On day 25, the mice are euthanized, omenta removed and RNA isolated to allow quantification of tumor burden as well as immune cell changes by qRT-PCR. Alternatively, lymphocytes may be isolated from the omenta using the “walkout” method and phenotyped by flow cytometry. Representative data shown are original unpublished data generated by the authors.

**Table 1 cells-10-03155-t001:** Summary of different in vivo models developed to study tumor-associated exosomes and their effects on T cells.

Type of Model	Type of Cancer	Patient Derived?	Effect of Exosomes Reported	Reference
Orthotopic Syngeneic	Melanoma	No	MicroRNA-dependent apoptosis of CD4 T cells increased tumor growth	[71]
	Melanoma	No	PD-L1-dependent inhibition of proliferation and tumor infiltration of TILs, increased tumor growth	[43]
	Breast Cancer	No	PD-L1-dependent inhibition of cytotoxic T cell activity and increase in tumor growth	[98]
Heterotopic Syngeneic	Breast Cancer	No	Decrease in CD4 and CD8 frequencies in lung, promotion of metastasis	[92]
	Prostate Cancer, Colorectal Cancer	No	Decrease in number, cytotoxicity and proliferation of CD8 T cells, increase in Tim-3 expression on T cells, increased tumor growth	[44]
	Non-small cell lung cancer (NSCLC)	No	PD-L1-dependent decrease in TILs, increase in tumor growth	[99]
Orthotopic Xenograft	Melanoma	No	PD-L1-dependent increase in tumor growth	[43]
Heterotopic Xenograft	Melanoma	Partly	Phosphatidylserine-dependent increase in tumor growth	[10]
	Ovarian Cancer	Yes	Phosphatidylserine-dependent increase in tumor burden, decrease in T cell activation and proliferation	[10]

## Data Availability

Not applicable.

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
