# Peer review of "Tumor-Associated Exosomes: A Potential Therapeutic Target for Restoring Anti-Tumor T Cell Responses in Human Tumor Microenvironments"

_cells, 2021, doi:10.3390/cells10113155_

Round 1

Reviewer 1 Report

The role of tumor-Derived Exosomes in Immunosuppression and Immunotherapy is a hot topic in cancer research and has been extensively reviewed recently (Journal of Immunology Research, Volume 2020, Article ID 6272498, https://doi.org/10.1155/2020/6272498). This review focus on exosomes suppression of T cell function in the human tumor microenvironments. The review discusss exosomes biogenesis, isolation and characterization, the immunosuppressive effect on T cells, as well as the mechanisms associated with exosome-mediated suppression of T cell function. Furthermore, the authors highlight the Xenomimetic mouse (X-mouse) model as well as the Omental Tumor Xenograft (OTX) model that can be applied to evaluate the efficacy of a novel molecule for targeting immunosuppressive tumor-associated exosomes.

Overall, this is is a well written, easy to follow review. The authors may wish to consider a few comments:

  1. It has been suggested that exosomal DNA and RNA may play an important role in genetic communication between different cells. These should be included in figure2.
  2. Exosomes are involved in pre-metastasis niche formation, are these also related to local T cells function suppression?

Minor comments

  1. Abbreviations: phosphatidylserine (PS) was repeatedly abbreviated.
  2. Please define TCR at the first-time use (page 4 line 154)

Reviewer 2 Report

Shenoy et al. reviewed the potential of tumour-associated exosomes in therapeutic targets for restoring anti-tumour T cell responses in tumour-microenvironments. 

The review is well organised and well written. 

Minor comments.

Exosome Biogenesis, Isolation and Characterization subheading requires more details. especially the Isolation of exosomes. 

I don't think so you can add unpublished original data?

Table 1. adding mechanistic information will improve the manuscript. 
